# Dexamethasone-Induced Insulin Resistance Attenuation by Oral Sulfur–Oxidovanadium(IV) Complex Treatment in Mice

**DOI:** 10.3390/ph17060760

**Published:** 2024-06-10

**Authors:** Eucilene K. Batista, Lidiane M. A. de Lima, Dayane A. Gomes, Debbie C. Crans, Wagner E. Silva, Mônica F. Belian, Eduardo C. Lira

**Affiliations:** 1Departamento de Fisiologia e Farmacologia, Centro de Biociências, Universidade Federal de Pernambuco, Recife 50670-901, PE, Brazil; kelitabl@gmail.com (E.K.B.); dayane.gomes@ufpe.br (D.A.G.); eduardo.carvalholira@ufpe.br (E.C.L.); 2Departamento de Química, Universidade Federal Rural de Pernambuco, Recife 52171-900, PE, Brazil; lidiane.lima@ufrpe.br (L.M.A.d.L.); wesdqr@gmail.com (W.E.S.); 3Department of Chemistry, Colorado State University, Fort Collins, CO 80513, USA; 4Cell and Molecular Biology Program, Colorado State University, Fort Collins, CO 80513, USA

**Keywords:** diabetes *mellitus*, vanadium complex, insulin resistance

## Abstract

Vanadium compounds are known to exert insulin-enhancing activity, normalize elevated blood glucose levels in diabetic subjects, and show significant activity in models of insulin resistance (IR). Faced with insulin resistance, the present work investigates the antidiabetic performance of a known oxidovanadium(IV)-based coordination compound—[V^IV^O(octd)]—and effects associated with glucocorticoid-induced insulin resistance in mice. The effects of [V^IV^O(octd)] were evaluated in a female Swiss mice model of insulin resistance induced by seven days of dexamethasone treatment in comparison with groups receiving metformin treatment. Biological assays such as hematological, TyG index, hepatic lipids, glycogen, oxidative stress in the liver, and oral glucose tolerance tests were evaluated. [V^IV^O(octd)] was characterized with ^51^V NMR, infrared spectroscopy (FTIR), electron paramagnetic resonance (EPR), electronic absorption spectroscopy, and mass spectrometry (ESI–FT–MS). The [V^IV^O(octd)] oral treatment (50 mg/kg) had an antioxidant effect, reducing 50% of fast blood glucose (*p* < 0.05) and 25% of the TyG index, which is used to estimate insulin resistance (*p* < 0.05), compared with the non-treated group. The oxidovanadium–sulfur compound is a promising antihyperglycemic therapeutic, including in cases aggravated by insulin resistance induced by glucocorticoid treatment.

## 1. Introduction

Vanadium-based compounds are known to exert insulin-enhancing activity and have demonstrated their antidiabetic activity in vitro and in vivo [1,2,3,4]. Vanadium complexes demonstrate superior biological effects in contrast to salts. Specifically, bis(ethylmaltolato)oxidovanadium(IV) (BEOV) exhibited promising outcomes during Phase 1 and Phase 2 clinical trials; however, the compound was discontinued due to renal toxicity concerns [5]. There is therefore a need to reduce toxicity and determine some of the beneficial effects of vanadium compounds.

Furthermore, vanadium-based compounds have biological effects on various aspects of carbohydrate and lipid metabolism, including glucose uptake, glycolysis and glycolytic enzymes, glucose oxidation, glucose production, glycogen synthesis, and lipogenesis, and they show significant activity in models of insulin resistance [6]. The potential for V compounds to overcome insulin resistance could be considerable and is investigated in further detail in this work.

The animal model employed for diabetes *mellitus* (DM) to investigate the vanadium compound effects is the STZ type I model—caused by decreasing and absent insulin secretion by pancreatic beta cells [5,6]. Nonetheless, DM2 constitutes approximately 95% of all cases and is distinguished by enduring insulin resistance (IR), leading to disruptions in protein, lipid, and carbohydrate metabolism [7]. Animal models with high-fat diets and glucocorticoid administration have been used to test for insulin resistance induced in rodent subjects [8].

Glucocorticoids (GCs) are steroid hormones produced and released by the adrenal cortex under hypothalamic–pituitary–adrenal axis regulation. They interact with the glucocorticoid receptor (GR), modulating several physiological functions [8]. However, excessive levels of GCs can lead to hypertensive conditions, increased body fat mass, osteoporosis, depression, infection risk, muscle wasting, and hyperglycemia [9].

The effects of GCs on glucose and lipid metabolism involve several mechanisms, such as insulin secretion inhibition from pancreatic beta cells, a reduction in glucose utilization, lipolysis, skeletal muscle proteolysis, and hepatic glucose production [10]. In addition, the clinical use of GCs is typically associated with glucose, protein, and lipid disturbances that are reproducible in both rodents and humans [11,12,13,14].

Dexamethasone (DEX) is a non-selective synthetic glucocorticoid widely prescribed because of its anti-inflammatory, anti-allergic, and immunosuppressive properties. It is approximately 50 to 100-fold more potent than cortisol. However, when administered in excess, DEX induces side effects such as muscle catabolism, hyperphagia, increased adiposity, dyslipidemia and hypertriacylglycerolemia, and IR in vitro and in vivo [10,15,16,17,18,19,20]. In addition, the long-term administration of DEX leads to the generation of free radicals such as superoxide, hydrogen peroxide, and hydroxyl radicals, which contribute to oxidative stress and deteriorate both insulin action and secretion, accelerating the consequent appearance of type 2 DM [21]. The adverse metabolic effects of DEX treatment may be reversible upon rapid discontinuation [22]. However, patients receiving GCs are often subjected to prolonged therapy, which may result in irreversible metabolic damage, such as IR, oxidative stress, and type 2 diabetes [23]. Considering the diabetogenic effect of GC therapy, it is important to propose new therapies to prevent undesirable side effects of the acute and long-term use of synthetic GCs such as dexamethasone.

In recent years, GC-induced disorders have gained prominence due to the need for emergency clinical use during the COVID-19 pandemic. Glucocorticoids, especially DEX, were widely used in patients with severe acute respiratory syndrome (SARS) to interrupt the inflammatory cascades caused by the virus [24,25]. For these cases, the use of some oral medications showed a low efficiency in post-COVID glycemic control. Despite this, new alternatives for glycemic control in these patients have been studied using an insulin-resistant diabetic animal model induced via dexamethasone [26]. In this context, oxidovanadium complexes have emerged as an efficient therapeutic proposal capable of controlling metabolic disorders associated with DM [1,2,3,4].

Recently, we evaluated the therapeutic potential of the oxidovanadium(IV) complex [V^IV^O(octd)], containing a 3,6-dithio-1,8-octanediol ligand (octd), which was evaluated as an oral antidiabetic agent on STZ-induced diabetic Wistar rats [27]. The proposed oxidovanadium complex has not been subjected to biological studies probing the effects associated with IR factors induced by acute exposure to dexamethasone. In this study, the biological effects of the [V^IV^O(octd)] complex on dexamethasone-induced insulin resistance in female mice were evaluated.

## 2. Results

### 2.1. Chemical Characterization of [V^IV^O(octd)] in Solution

The [V^IV^O(octd)] complex, a dark green solid with the chemical formula [C_6_H_12_O_3_S_2_V], was previously synthesized and characterized as described by Lima et al. (2023) [27]. Its mass spectrum shows a peak at *m*/*z* = 246.96599 corresponding to the molecular ion [M] (C_6_H_12_O_3_S_2_V)^+^. The FTIR spectrum of the solid contains characteristic bands of the oxidovanadium complex at 961, 615, and 530 cm^−1^ assigned to ν(V=O), ν(C–S), and ν(V-O) stretching, respectively.

Additionally, a time-dependent decomposition study of [V^IV^O(octd)] in an aqueous solution was realized using ^51^V NMR, EPR, and electronic absorption spectroscopy, thus enabling the monitoring of vanadium(V) species formation [27]. The EPR spectrum of [V^IV^O(octd)] in DMSO displayed an eight-line pattern (Appendix A), consistent with the vanadium nuclear hyperfine couplings of oxidovanadium(IV) complexes (^51^V, I = 7/2) and like V^IV^OSO_4_, which is used as standard.

The ^51^V NMR of a 2.0 mmol L^−1^ aqueous solution of [V^IV^O(octd)], pH~4, showed a signal at –557 ppm assigned to a monomer V_1_ (H_2_VO_4_^–^). Over time, after 72 h, the ^51^V NMR showed signals (–424, –504, and –522 ppm) assigned to the oligomer decavanadate V_10_ (V_10_O_28_^6−^) (Appendix A).

The electronic absorption spectroscopy of the 2.0 mmol L^−1^ aqueous solution of the [V^IV^O(octd)] complex showed a broad band that extended from 600 nm to the near-infrared (maximum in 996 nm) assigned to an intervalence charge transfer transition (IVCT, V^IV^ → V^V^) characteristic of a mixture of the V^IV^ and V^V^ complexes, partly [V^IV^O(octd)] in the reduced form and partly in the oxidized form [V^V^O_2_(octd)]. This hypothesis was supported by the ^51^V NMR and EPR tests [27]; after 1 h, the V(IV) compound was oxidized to the V^V^ species, which was consistent with the decrease in absorbance observed in the absorption spectrum over time (Figure 1).

Hence, these studies show [V^IV^O(octd)] slowly decomposes in an aqueous solution, resulting in the oxidation of [V^IV^O(octd)] in the oxidovanadium(V) complex, [V^V^O_2_(octd)^–^]. These last results confirmed the oxidovanadium(IV) speciation dynamics and its chronological conversion to vanadium(V)-analogous complex, as illustrated in Figure 1.

### 2.2. Dexamethasone-Induced Insulin Resistance in Mice

The scientific literature already described that the sulfur–oxidovanadium(IV) complex—[V^IV^O(octd)]—reduces hyperglycemia and lipid profiles in type I diabetic STZ-induced rats [27]. The current study highlights that the used animal model (mice) treated with predetermined doses (1 mg kg^−1^, intraperitoneal, 7 days) of dexamethasone showed significant elevations of their fasting hyperglycemia (9.42 ± 0.50 vs. 3.64 ± 0.08 of the control group, *p* < 0.05) and hypertriglyceridemia (2.46 ± 0.19 vs. 0.94 ± 0.09 of the control group, *p* < 0.05, Figure 2).

The treatment based on [V^IV^O(octd)] reduced hyperglycemia (50%, *p* < 0.05) in both assayed doses and reduced hypertriglyceridemia in a dose-dependent manner (35 and 50%). Accordingly, we used another homeostatic model based on the TYG index to estimate peripheral insulin sensitivity [28]. It was observed that the TyG index was increased (25%, *p* < 0.05) by the DEXA administration, confirming the insulin resistance in the mice. This effect was reduced by the [V^IV^O(octd)] treatment in a dose-dependent manner (10 and 25%, *p* < 0.05). Another important aspect was the metformin treatment, which needed higher doses to produce similar results to [V^IV^O(octd)].

There were no significant differences in body weight between animals of the control, standard (DEXA Met), and tested (V^IV^ complex) groups. As shown in Figure 3, DEXA administration slightly reduced total body weight gain and total food intake (~10%, *p* < 0.05) without any alteration in fluid intake.

Neither oral [V^IV^O(octd)] administration nor metformin treatment significantly affected any glucocorticoid-induced changes in the characteristics listed in Table 1.

As expected, DEXA reduced the adrenal gland weight, demonstrating the effectiveness of this synthetic glucocorticoid administration (Table 2). The oral coordination compound treatment did not affect this last dexamethasone-induced change in mice. Heart, liver, and kidney were not physiologically and morphologically influenced by DEXA and [V^IV^O(octd)] treatment.

On the other hand, dexamethasone-induced skeletal muscle atrophy in the tibial anterior (25%, *p* < 0.05) and soleus (45%, *p* < 0.05) was prevented in the tibial anterior and attenuated in the soleus muscles (50%, *p* < 0.05) by both doses of [V^IV^O(octd)]. However, DEXA did not change abdominal adipose tissue weight; this synthetic glucocorticoid increased RETRO adipose tissue weight (70%, *p* < 0.05). The higher dose of [V^IV^O(octd)] blocked the effect of DEXA on hypertrophy in RETRO adipose tissue (Table 2).

As expected, the metformin-treated mice showed an increased adipose tissue weight compared with the control; however, its action on the dexamethasone-induced mice did not have a significant hypertrophic RETRO adipose tissue effect.

Insulin resistance is usually accompanied by glucose intolerance. The DEXA group had higher basal glucose levels than the control group (143.8 ± 3.3 vs. 80.1 ± 5.3 of control, *p* < 0.05, Figure 4). As expected, the synthetic glucocorticoid (DEXA) administration induced oral glucose intolerance in mice, as seen in the AUC (Figure 4). Nonetheless, only the [V^IV^O(octd)] treatment at a higher dose improved oral glucose tolerance in the DEXA-induced mice.

### 2.3. Serum Profile

Nowadays, clinical endocrinology determines that higher glucose levels, in human blood serum, produce a spontaneous glycation reaction, enabling stable Amadori products. In this way, it was observed that the dexamethasone treatment induced an increase in animal serum fructosamine levels (2.4-fold, *p* < 0.05) in opposition to an advanced glycation end-product effect that was prevented by the [V^IV^O(octd)] treatment. Regarding serum lipid profiles, the synthetic glucocorticoid increased the total cholesterol (~70%, *p* < 0.05) and non-HDL cholesterol (5.3-fold, *p* < 0.05) but reduced C-HDL (~50%, *p* < 0.05). Dexamethasone elevated the calculated atherogenic index (4.0-fold, *p* < 0.05). The [V^IV^O(octd)] treatment reduced the total and non-HDL cholesterol in the DEXA-treated mice. However, the [V^IV^O(octd)] treatment did not improve the C-HDL levels. In addition, the atherogenic index was decreased by the [V^IV^O(octd)] treatment in a dose-dependent manner. In addition, the DEXA treatment increased serum glycerol levels (~90%, *p* < 0.05), which were blocked by both treatments based on [V^IV^O(octd)] and metformin (Table 3).

DEXA has been associated with serum transaminase level elevations. In this work, it was verified that dexamethasone increased the alanine aminotransferase (65%, *p* < 0.05) and aspartate aminotransferase levels (90%, *p* < 0.05) without any change in the total protein content (Table 3). The [V^IV^O(octd)] treatment did not change the glucocorticoid-induced increase in AST, but it did reduce ALT levels (25%, *p* < 0.05). Similarly, metformin only reduced ALT serum contents (50%, *p* < 0.05).

### 2.4. Liver Lipid Profile

Glucocorticoids influence lipid metabolism in several organs, including the liver; here, the high dosage administration of dexamethasone disturbed the hepatic metabolism of lipids. It was observed that the DEXA-treated mice showed an increase in total lipid (8.7-fold, *p* < 0.05), total cholesterol (TC, 2.4-fold, *p* < 0.05), and triglyceride (TG, 2.4-fold, *p* < 0.05) accumulation but reduced C-HDL (50%, *p* < 0.05) in the liver. The [V^IV^O(octd)] treatment reduced the total lipid content in the liver (85%, *p* < 0.05), as observed in the TG case, whose reduction occurred in a dose-dependent manner. Fasting mice treated with dexamethasone showed an increase in hepatic glycogen (12-fold). However, the [V^IV^O(octd)] treatment also reduced the DEXA-induced storage of liver glycogen in a dose-dependent manner (Table 4).

### 2.5. Oxidative Stress

The antihyperglycemic mechanism of the [V^IV^O(octd)] was investigated through the analysis of the malondialdehyde (MDA) and glutathione concentrations; and enzymatic activities of the superoxide dismutase (SOD) and catalase (CAT). DEXA administration caused significant increases in liver nitrite (30%, *p* < 0.05) and MDA (90%, *p* < 0.05) levels, which was accompanied by a reduction in the GSH/GSSH ratio (25%, *p* < 0.05). The [V^IV^O(octd)] treatment reduced MDA levels, nitrite contents, and the GSH/GSSG ratio. In addition, the synthetic glucocorticoid treatment caused a reduction in CAT but not in SOD activity, similar behavior to both doses of the [V^IV^O(octd)] and metformin treatment (Table 5).

## 3. Discussion

Recently, Lima and collaborators (2023) reported antidiabetic effects in STZ-induced diabetic rats of the sulfur–oxidovanadium(IV) complex—[V^IV^O(octd)] [27]. In the current study, in addition to demonstrating the antioxidant properties of the [V^IV^O(octd)] complex was possible to confirm the fasting blood glucose levels reduction, as well as attenuate the metabolic disturbances associated with insulin resistance (IR), preserving the lean mass, and reduced fat depots in female mice. [V^IV^O(octd)] complex also reduced the serum lipid profile and the hepatic lipid accumulation in DEXA-treated mice. 

Since the 1970s, several oxidovanadium complexes have been discussed as insulin-enhancer agents with beneficial effects on glucose and lipid metabolism, in vitro and in vivo [1,2,3,4,5,6,28,29,30,31]. Among the various animal models of diabetes, the most widely used for vanadium compounds is streptozotocin (STZ) induced [32,33,34,35,36]. However, the STZ-induced DM model more closely resembles type I diabetes, and type II diabetes corresponds to around 95% of the cases. In addition, type 2 DM is characterized by peripherical IR, mainly in the liver, white adipose, and skeletal muscle tissues. There have been a few papers related to the investigation of vanadium compounds using IR models, high-fat diets, and synthetic glucocorticoids, which are the conditions that most represent type 2 diabetic patients [19,20].

In this way, the present work verified the influence of the administration of an oxidovanadium complex, [V^IV^O(octd)], on an animal model that consisted of mice with dexamethasone-induced IR, and it evaluated the complex’s potential therapeutical use as a novel antihyperglycemic agent. The dexamethasone-treated mice probably exhibited a reduction in corticosterone values due to the HPA axis inhibition, as suggested by adrenal gland weight (Table 2). These results confirm that excess dexamethasone can induce energetic metabolism damage such as hyperglycemia, hyper-triacylglyceridemia, and IR in rodents [10,18]. In the current research, we used metabolic parameterization methods such as the TyG index, a validated indicator to demonstrate IR that has been compared with the gold standard technique of the “euglycemic–hyperinsulinemic” clamp test. It is well known in the literature that the administration of glucocorticoids induces a decrease in peripheral glucose disposal and increased liver gluconeogenesis, which is associated with damage in the insulin anabolic action [9].

Interestingly, the [V^IV^O(octd)] use reduced fasting glucose and TG levels, as well as calculated TyG index (Figure 2), all for the animal model assayed here. Nonetheless, Barbera et al. (2001) [20] showed that a treatment strategy based on the VOSO_4_ did not block or attenuate DEXA-induced IR in rats. On the other hand, therapeutical strategies based on vanadium complexes are promising; for example, the use of the bis(alpha-furancarboxylato)oxidovanadium(IV) (BFOV) enhanced the action of insulin and completely prevented the development of insulin resistance induced by dexamethasone, modulating the gene expression of key proteins of the insulin cascade such as insulin receptor substrate 1 (IRS-1) and glucose transporter 4 (GLUT4) in 3T3-L1 adipocytes [19]. Although we have not yet clarified the molecular mechanism involved in the anti-IR of [V^IV^O(octd)] complex, it is reasonable to hypothesize that this vanadium-based compound activates the insulin pathway in metabolically important tissues such as the liver and skeletal muscle, which counteracts the anti-insulin effects of dexamethasone.

Based on previous speciation studies, it was possible to predict the real V species responsible for the biological effects described in this study [28]. [V^(IV)^O(octd)] is the major species at pH = 4; then, it undergoes partial speciation. On the other hand, at pH = 7 in an aqueous solution, the initial compound is less stable because hydrolysis/decomposition occurs from 0 h, originating the oxidation product of the [V^V^O_2_(octd)^–^] complex, ligands and metals free, that are oxidized to the V^(V)^ species, V_1_, V_2_, V_4,_ and V_5_, at pH = 7 and 24 h. Species of the biological system responsible for blood glucose and insulin-resistance reduction depend on time. Therefore, it is reasonable to hypothesize that [V^IV^O(octd)] and its oxidation/decomposition products affect the antihyperglycemic and could activate the insulin pathway in metabolically important tissues such as the liver, adipose tissue, and skeletal muscle, which counteracts the anti-insulin action of DEXA.

Dexamethasone administration generates decreases in both body weight gain and food intake, as illustrated in Table 1. The observed decline in energy intake does not appear to be the primary determinant for reduced body weight gain in rats subjected to exogenous glucocorticoid treatment [9,10,37]. It is well described in the literature that GC treatment increases plasma insulin and leptin levels, hormones involved in the anorexigenic response in the hypothalamus [37,38]. Nevertheless, excess glucocorticoids produce a negative nitrogen balance, which may contribute to body weight loss [39,40]. Aru et al. (2018) showed similar results with a DEXA-induced reduction in the weight of fast-twitch muscle without any alteration in the weight of slow-twitch muscle [41]. The [V^IV^O(octd)] administration counteracted the catabolic effect in skeletal muscle caused by dexamethasone in mice. Also, the proposed vanadium-based compound spared skeletal muscle protein, which may explain, at least in part, the body weight gain in mice treated with [V^IV^O(octd)] complex. This effect resulted in an adsorption improvement of insulin, that was induced by the proposed oxidovanadium complex.

It is well established that glucocorticoids promote adiposity, triglyceride synthesis, and adipose tissue hypertrophy [18,22,26]. These aspects were reinforced in this work, whose results showed that retroperitoneal adipose tissue (RETRO) weight increased after 14 days of exposure to DEXA (Table 2). The effect of glucocorticoids on adipose tissue is not entirely understood and is still controversial. For example, Aru et al. (2018) reported reduced fat body mass in 22-month-old rats overexposed to dexamethasone for 10 days [41]. On the other hand, other research has shown that synthetic glucocorticoids induce adipose tissue gain [18,22,42]. Ferreira et al. (2017) showed that despite impairing insulin-stimulated glucose uptake in both RETRO and epidydimal adipose tissue, dexamethasone increased adipogenesis, glyceroneogenesis, and phosphoenolpyruvate carboxykinase (PEPCK-C) activity in RETRO tissue, accompanied by a reduction in AKT phosphorylation, which suggests IR [18]. Concerning the [V^IV^O(octd)] complex, which blocked the adipogenic effect of DEXA in RETRO adipose tissue, the current proposed therapeutic prototype did not present the usual undesirable effects of antidiabetic pharmacotherapy such as adipose tissue gain and lipodystrophy.

The vanadium complex may contribute to differentially reestablishing insulin sensitivity among body tissues to reduce blood glucose levels with concomitant adipogenesis reduction. GCs may also increase lipolysis ex vivo and in vivo, which may be detected by increased glycerol levels, associated with IR [18]. Glycerol is a lipolysis-derived substrate that may be used for glucose hepatic production [43]. Accordingly, [V^IV^O(octd)] reduced the availability of this gluconeogenesis substrate (Table 3), which could justify the reduction in hyperglycemia in the DEXA-treated mice. In other words, this sulfur–oxidovanadium compound may counteract the hyperglycemic side effect of dexamethasone treatment because it reduces the gluconeogenic substrates, such as amino acids and glycerol, in the liver.

It has been demonstrated that glucocorticoids stimulate the storage of glucose as glycogen in the liver, including in fasting conditions, and it has been associated with GC-stimulated glycogen synthase (GS) and/or the inhibition of glycogen phosphorylase (GP) activity [44]. Fasting-induced depletion in liver glycogen levels in adrenalectomized rats is blocked by dexamethasone administration [45]. Furthermore, Praestholm et al. (2021) demonstrated that activating intracellular glucocorticoid receptors (GRs) is essential to hepatic glucose uptake and liver glycogen storage [46]. In agreement with this, GRs directly regulate the expression of glucose kinase, a key enzyme to glucose utilization and storage in the liver. In addition, insulin and glucocorticoids have a synergic effect on GS activity and glycogen content in different physiological conditions [46]. Previous research has demonstrated that DEXA increases hepatic glycogen [22]. Our results showed that [V^IV^O(octd)] and metformin, an insulin-sensitizing drug used as a control, reduced the glycogen storage induced by DEXA in fasting mice. Considering that DEXA induces hyperinsulinemia in mice [22,42], the present vanadium-based compound attenuated the effect of GCs on glycogen storage. It is possible that [V^IV^O(octd)] improves insulin sensitivity in the liver and improves the glycogen turnover in DEXA-treated mice.

It is noteworthy that while GCs induce IR in the gluconeogenic pathway by upregulating the gene expression of key enzymes, such as phosphoenolpyruvate carboxykinase (PEPCK) and glucose-6-phosphatase (G6-P), the lipogenic effect of insulin remains preserved [26]. Compensatory hyperinsulinemia occurs in response to GC-induced high blood glucose levels, which stimulate de novo lipogenesis (DNL) and subsequent lipid accumulation in the liver. In the current research, impairments in serum lipid profiles were associated with the atherogenic index increase in GC-treated mice (Table 3). Furthermore, the GC treatment (DEXA) caused the imbalance of hepatic lipid homeostasis between the acquisition and removal of TG/fatty acid (Table 4). The liver lipids machinery (synthesis and export) for the other tissues is complex and involves several key enzymes for lipid uptake, DNL, fatty acid β-oxidation, and liver lipid export [26].

In this regard, GC action is not only affected by circulating serum levels or densities of GRs but also by tissue-specific GC-activating enzyme 11 β-hydroxysteroid dehydrogenase type 1 (11 β-HSD-1) or deactivating enzyme 11 β-hydroxysteroid dehydrogenase type 2 (11 β-HSD-2) [8,47]. In the present study, [V^IV^O(octd)] administration reduced the effects of dexamethasone on lipid accumulation in the liver and the serum lipid profile; previously, 11β-HSD-1 knockout mice ameliorated glucose tolerance and reduced gluconeogenic gene expression during fasting [47]. On the other hand, the overexpression of this limiting enzyme for activating GCs in adipose tissue was shown to cause glucose intolerance, IR, and moderate obesity in mice [48]. However, we did not evaluate the expression of GRs and 11β-HSD-1, so it is possible to suggest that [V^IV^O(octd)] complex not only improved insulin resistance by the classical mechanisms but also probably modified the GC signaling and availability for tissues. This hypothesis becomes even more attractive when we compare the effects of [V^IV^O(octd)] complex on DM1 rats [27] and dexamethasone-induced IR mice.

One of the most important reasons for the undesirable effects of treatment using DEXA is that it induces the overproduction of reactive nitrogen (RNS) and oxygen species (ROS), causing redox imbalance and leading to cellular damage [49]. DEXA promoted an increase in oxidative stress biomarkers, which was counteracted by the administration of the sulfur–oxidovanadium(IV) complex (Table 5). The antioxidant effect of some vanadium compounds is still controversial. Some vanadium-based compounds exacerbate allergic airway inflammation by triggering reactive oxidative stress [50]. Polyoxovanadate, for example, induces severe toxicity in mice, at least in part by increasing oxidative stress, as seen in both an elevation of MDA levels and a reduction in the GSH/GSSG ratio [51]. The design of different ligands in the coordination sphere of vanadium-based complexes has allowed for more bioactive compound building (antioxidant, hepato-protective, and antidiabetic) with reduced side effects. This enables their therapeutic uses in several diseases, including cardiovascular diseases (CVDs), obesity, and IR [52]. The data suggest that [V^IV^O(octd)] may modulate the cellular enzymatic apparatus to protect against the oxidative stress induced by synthetic GCs such as dexamethasone.

In addition, a primary consequence of a high blood glucose level is a non-enzymatic glycation reaction, which produces advanced-glycation end products (AGEs) and induces glucose-oxidative damage. Serum fructosamine is a glycated protein that reflects average glycemia over the previous 2–3 weeks [53]. Here, dexamethasone elevated glucose-oxidative stress, as seen in the fructosamine higher levels in mice. However, the sulfur–oxidovanadium(IV) complex treatment reduced this AGE production. These results confirmed that the proposed vanadium-based compound increased glucose metabolism and prevented glucose-oxidative stress in IR mice.

Modifications of the structures of vanadium compounds have been used to reduce their toxicity and increase their chemical stability and bioavailability in living organisms. In this case, an ionophore ligand, with an S_2_O_2_ coordination mode, could contribute to the biological effects reported in this study. However, biodistribution and biotransformation studies of [V^IV^O(octd)] must still be conducted; this compound has no toxicity in Wistar rats [27]. In the future, it will be promising to use [V^IV^O(octd)] as a potential therapeutical agent to protect patients against the metabolic side effects of synthetic GCs, mainly dyslipidemia, oxidative stress, and IR.

## 4. Materials and Methods

### 4.1. Chemicals and General Methods

1,8-octenodiol 97%, sodium methoxide, and oxidovanadium(IV) sulfate hydrate (97%), all reagents from Sigma-Aldrich^®^ (St. Louis, MO, USA), were used without further purification. Deionized water (18.2 MΩ cm) was used to synthesize and characterize the vanadium complex. ^51^V NMR data were collected on a Bruker 400.13 MHz spectrometer (Billerica, MA, USA), with a 78.9 MHz frequency for vanadium at 298 K using 4096 scans and a spectra width from −53 to −1043 ppm, referenced to an external VOCl_3_ standard (0.00 ppm). Fourier transform infrared (IR) spectra (400 to 4000 cm^–1^) were recorded in the ATR mode on a Varian spectrophotometer (Santa Clara, CA, USA) with a resolution of 4 cm^–1^. Electron paramagnetic resonance in X-band (EPR) (9.84 GHz) was recorded at 77 K on a Bruker X-band (Billerica, MA, USA), 9.5 GHz. An aqueous solution of V^IV^OSO_4_ (50 mmol L^−1^) was used as a standard. The electronic absorption spectra of 2mmol L^−1^ vanadium complex were observed from 300 to 1100 nm using a Shimadzu UV–Vis–NIR Spectrometer (Kyoto, Japan). The ESI–FT–MS data were collected with an Exactive Plus mass spectrometer, positive mode data (Thermo Scientific, Bremen, Germany) [27].

### 4.2. Synthesis of the Sulfur–Oxidovanadium(IV) Complex—[V^IV^O(octd)]

The synthesis was reproduced according to the procedure described by Lima et al. (2023) [27]. In a round-bottom flask, 3,6-dithio-1,8-octanediol 97% (OCTD) (0.27 g, 1.5 mmol) and sodium methoxide (0.16 g, 3.0 mmol) were combined in 15.0 mL of methanol. The mixture was stirred at 25 °C for 1 h. Following this, VOSO_4_.5H_2_O (0.24 g, 1.5 mmol), previously dissolved in 5 mL of deionized water, was introduced into the reaction mixture and stirred at 25 °C for 12 h. The resulting solid was then vacuum filtered, washed with ethanol and diethyl ether, and dried under high vacuum conditions (Yield: 89%).

### 4.3. Animals and Treatment

Female Swiss mice of about two months old and weighing 35 ± 3 g were maintained in sanitized polypropylene cages (5 per cage) under standard conditions of temperature (23 ± 2 °C), relative humidity (55 ± 5%), and photoperiod (12 h light and 12 h dark) with free access to food and water. All animal care procedures were carried out according to the National Institutes of Health Guide for the Care and Use of Laboratory Animals (NIH Publication 8023, revised 1978), and they were approved by the Ethics Committee on the Use of Animals of the Federal University of Pernambuco (CEUA/UFPE, protocol 0052/2020).

Insulin resistance was induced in female mice (n = 8–10 per group) by dexamethasone dose administration (DEXA, i.p., 1 mg kg^−1^ body weight) for 7 consecutive days before they were divided into 4 groups: (a) the DEXA group continued, receiving only DEXA for next 7 days; (b) V_25_ and (c) V_50_ were only treated with DEXA for first 7 days and then received doses (oral gavage) of 25 and 50 mg kg^−1^ of sulfur–oxidovanadium(IV), respectively, for next 7 days. The control group received only saline 0.9% during 14 days of treatment (1 mL/kg, o.v.).

### 4.4. Biochemical Analysis

On the 14th day after treatment, fasted mice were anesthetized, and blood was drawn via the retro-orbital technique with or without heparin for both plasma and serum biochemical analyses. Between the 7th and 14th days, were realized triacylglycerol and glucose assays to calculate the TyG index [54]. In addition, the serum levels of albumin (ALB), alanine aminotransferase (ALT), aspartate aminotransferase (AST), glycerol, total cholesterol (TC), total protein (TP), fructosamine, and triacylglycerol (TG) were measured with kits from Labtest^®^ (Santa Lagoa, MG, Brazil) following the protocol of the manufacturer. The non- and high-density lipoprotein cholesterol (C-HDL) levels were calculated as (TC)—(C-HDL).

At the end of treatment, the total lipids were extracted from liver tissue (100 mg) according to Folch’s procedure [55]. In brief, the lipids were extracted by homogenizing the tissue, using a 2:1 chloroform-methanol (*v*/*v*) mixture, and filtering the homogenate. After that, the filter was mixed with saline 0.9%. The upper phase was aspirated, and an aliquot from the lower phase (500 µL) was transferred into a pre-weighed glass tube and dried overnight. Then, the remaining lipid content was weighed. The total liver cholesterol, C-HDL, and triglycerides were measured using enzymatic kits from Labtest ^®^ (Santa Lagoa, MG, Brazil), following the manufacturer’s protocol. After dissolving in potassium hydroxide, the hepatic glycogen was measured with the anthrone method [56].

### 4.5. Oxidative Stress Evaluation in the Liver

Aiming to evaluate some oxidative stress markers in the liver, after 14 days of the experiment, the animals were euthanized, and the liver was immediately excised and washed in an ice-cold Tris–HCl buffer (0.1 mol L^−1^, pH 7.4). After that, the samples were rinsed again in ice-cold 0.15 mol L^−1^ of potassium chloride and homogenized (10% *w*/*v*) using a 0.05% potassium dihydrogen phosphate buffer (pH 7.5) containing 1 mmol L^−1^ of EDTA, 1 mmol L^−1^ of sodium orthovanadate, and 200 µg/mL of phenylmethanesulfonyl fluoride (PMSF). The homogenates were centrifuged at 4000 rpm and 4 °C for 10 min. The supernatant was collected and used for enzymatic assays. The protein content was analyzed with the Branford method. Afterward, we conducted some assays, such as by measuring lipid peroxidation through the thiobarbituric acid (TBAR) reaction with malondialdehyde (MDA) according to Uchiyama and Mihara [57]. Glutathione peroxidase (GPx) activity was measured according to the method by Paglia and Valentine [58]. The reduced glutathione levels (GSH) measurement was carried out using the Hissin and Hilf method [59]. The antioxidant activity of the enzyme superoxide dismutase (SOD) was assessed according to the protocol developed by Misra and Fridovich [60], and catalase (CAT) activity was monitored according to Aebi [61].

### 4.6. Oral Glucose Tolerance Test (OGTT)

On the 14th day of the experiment, an OGTT was carried out to evaluate the glucose tolerance in the overnight fasted animals, as distributed in the following groups: non-diabetic control, dexamethasone (DEXA), vanadium-treated animals with doses of 25 mg kg^−1^ (V^IV^VO_25_) and 50 mg kg^−1^ (V^IV^VO_50_), and metformin-treated animals (MET). All animals received a dose of 2.5 g of glucose/kg (o.v.), and blood glucose was measured from blood samples withdrawn from the tip of the tail before glucose administration (t = 0) and after 15, 30, 60, 90, 120, 150, 180, and 210 min.

### 4.7. Statistical Analysis

The data were expressed as mean ± S.E.M. A one-way analysis of variance (ANOVA), followed by the Bonferroni test, was employed to analyze the data between treated groups and their respective control groups. A *p* < 0.05 value was considered statistically significant.

## 5. Conclusions

In the current study, an oxidovanadium–sulfur complex—[V^IV^O(octd)]—administration underwent partial speciation to become V(V), which demonstrated in vivo antihyperglycemic and lipid-lowering effects, probably by reducing dexamethasone-induced IR in mice. In addition, this vanadium complex demonstrated an in vivo antioxidant effect, which may prevent the development of metabolic diseases and their comorbidities. These results suggest that the oxidovanadium complex has emerged as an efficient therapeutic proposal capable of controlling the metabolic side effects associated with the therapeutic use of synthetic glucocorticoids. This becomes important when dexamethasone use is necessary, such as in transplant patients, and, more recently, for patients infected by the coronavirus with severe acute respiratory syndrome (SARS). For future research, it would be useful to evaluate the molecular level of proteins linked to insulin pathways, such as IR, IRS1, AKT, and glucocorticoid receptor (GR), to propose which metabolic pathways are activated by this vanadium complex.

## Data Availability

Data are contained within the article and Appendix A.

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
