# Peer review of "Dexamethasone-Induced Insulin Resistance Attenuation by Oral Sulfur–Oxidovanadium(IV) Complex Treatment in Mice"

_pharmaceuticals, 2024, doi:10.3390/ph17060760_

Round 1

Reviewer 1 Report

Comments and Suggestions for Authors

Manuscript titled “Dexamethasone-induced insulin resistance attenuation by oral sulfur-oxidovanadium(IV) complex treatment in mice ” by Batista et al. describes vanadium compounds exert insulin-enhancing activity, normalize elevated blood glucose levels in diabetic subjects, and show significant activity in models of insulin resistance (IR). It is an interesting work and well supported by the experimental data. The manuscript is well-written. Overall, the manuscript is well suited for Pharmaceuticals.  With that said, I recommend acceptance of the manuscript in its current form. 

Author Response

Journal Pharmaceuticals (ISSN 1424-8247)

Manuscript ID pharmaceuticals-3027971

Title: Dexamethasone-induced insulin resistance attenuation by oral sulfur-oxidovanadium(IV) complex treatment in mice

Special Issue: Recent Advances and Future Approaches in Preventive and Therapeutic Effects of Vanadium Complexes

RESPONSES TO REVIEWERS

We want to thank the reviewers for the opportunity to improve our manuscript.  We have done our best to respond to the reviewers and, in the following, we describe the step-by-step changes we have made to our manuscript.  The changes made in the manuscript are described below for the convenience of the reviewers and the editor.

Reviewer comments:

Reviewer #1: The manuscript titled “Dexamethasone-induced insulin resistance attenuation by oral sulfur-oxidovanadium(IV) complex treatment in mice ” by Batista et al. describes how vanadium compounds exert insulin-enhancing activity, normalize elevated blood glucose levels in diabetic subjects, and show significant activity in models of insulin resistance (IR). It is an interesting work and well supported by the experimental data. The manuscript is well-written. Overall, the manuscript is well-suited for Pharmaceuticals.  With that said, I recommend acceptance of the manuscript in its current form. 

Response: We want to thank the reviewer for recognizing the value of our manuscript.

Reviewer 2 Report

Comments and Suggestions for Authors

Manuscript Review Report:  Manuscript ID: pharmaceuticals-3027971

The manuscript describes the synthesis and characterization of V(IV) complex derived from 3,6-dithio-1,8-octanediol. The antidiabetic activity of the complex has been employed in vivo.

There are some comments that should be considered throughout the manuscript:

1.     The introduction should explain the difference between Type 1 and Type 2 diabetes and what type you plan to address. Furthermore, what features does the ligand have to be studied?

2.     Which solvent was used to measure the stability? Is it buffer solution, or another solvent?

3.     How did you confirm that the V(IV) was oxidized to V(V) by electronic spectra? If V(V) is present, so the peak should disappear, because it has no d-d transition peak.

4.     Cyclic voltammetry can be used to study the redox behavior of the complex.

5.     The discussion looks like an English summary, There are no interpretations in the discussion section; the results are only displayed. the data must be interpreted to reveal the real function of the V(IV) complex.

6.     Vanadium(IV) complex is applied as V(IV) and undergoes partial speciation to become V(V), which controls the amount of cellular intake of glucose. Could you provide a simplified scheme that illustrates the insulin-mimetic function of the V(IV) complex?

7.     Conclusion should supported by some obtained data.

Author Response

Journal Pharmaceuticals (ISSN 1424-8247)

Manuscript ID pharmaceuticals-3027971

Title: Dexamethasone-induced insulin resistance attenuation by oral sulfur-oxidovanadium(IV) complex treatment in mice

Special Issue: Recent Advances and Future Approaches in Preventive and Therapeutic Effects of Vanadium Complexes

RESPONSES TO REVIEWERS

We want to thank the reviewers for the opportunity to improve our manuscript.  We have done our best to respond to the reviewers and, in the following, we describe the step-by-step changes we have made to our manuscript.  The changes made in the manuscript are described below for the convenience of the reviewers and the editor.

Reviewer comments:

Reviewer #2: The manuscript describes the synthesis and characterization of the V(IV) complex derived from 3,6-dithio-1,8-octanediol. The antidiabetic activity of the complex has been employed in vivo.

 There are some comments that should be considered throughout the manuscript:

  1. The introduction should explain the difference between Type 1 and Type 2 diabetes and what type you plan to address. Furthermore, what features does the ligand have to be studied?

Response: We have now included the type of DM (page 2, lines 47-52). Aiming was to evaluate the oxidovanadium complex activity in an insulin resistance model induced by dexamethasone acute treatment in mice. We have now clarified our introduction to be more assertive.

  1. Which solvent was used to measure the stability? Is it buffer solution, or another solvent?

Response: The test was carried out in an aqueous solution to ensure the administration of the complex to the group of animals without changes to the species tested. As a result, the solution to be used was prepared within 15 min, between preparation and administration. The EPR and 51V NMR in an aqueous solution were able to demonstrate what the active species responsible for the antihyperglycemic effects observed in the previous study published by our group (reference 27) using STZ-induced rats treated with [VO(octd]) complex. Also, based on other papers that did not use a physiological buffer, we chose to give the solution orally to the animals in deionized water.

  1. How did you confirm that the V(IV) was oxidized to V(V) by electronic spectra? If V(V) is present, so the peak should disappear, because it has no d-d transition peak.

Response: The text was modified, as Figure 1 demonstrates a hypothesis of oxidation, not a confirmation. This hypothesis was supported by the 51V NMR and EPR tests, including in the Supplementary section and reference [27], after 1h, the V(IV) compound is oxidized to V(V) species, which is consistent with the decrease in absorbance observed in the absorption spectrum. Thank you for making us aware so we could clarify this point and improve this discussion in the section.

  1. Cyclic voltammetry can be used to study the redox behavior of the complex.

Response: Cyclic voltammetry is an important technique for studying the redox behavior of the complex, we consider that the results presented here and demonstrated in our previously published study [27], are sufficient to indicate that the [VO(octd)]complex undergoes partial speciation to become V(V).

  1. The discussion looks like an English summary, there are no interpretations in the discussion section; the results are only displayed. The data must be interpreted to reveal the real function of the V(IV) complex.

Response: We improved the discussion as recommended.

  1. Vanadium(IV) complex is applied as V(IV) and undergoes partial speciation to become V(V), which controls the amount of cellular intake of glucose. Could you provide a simplified scheme that illustrates the insulin-mimetic function of the V(IV) complex?

Response: We feel that it would not be appropriate to present this scheme again since it has already been described in another article done by our research group (Figure 10, reference: Lima, L. M. A.; da Silva, Amanda K.J.P.F.; Batista, E. K.; Postal, K.; Kostenkova, K.; Fenton, A.; Crans, D.C.; Silva, W.E.; Belian, M.F.; Lira, E. C. The antihyperglycemic and hypolipidemic activities of a sulfur-oxidovanadium(IV) complex. J Inorg Biochem. 2023, 241, 112127.)

However, to address this valuable suggestion, we have added some points about its mechanism of action in the discussion section (pages 9 and 10, lines 334-345).

  1. Conclusion should be supported by some obtained data.

Response: Improvements were made in the conclusion of the manuscript.

Reviewer 3 Report

Comments and Suggestions for Authors

Steps are needed to detail the reasons for the selection of the sulfur-oxovanadium(IV) complex and its advantages over existing treatments.

Over-representation of references over 10 years

In the “Materials and Methods” section, it is suggested to increase the detailed description of each step, especially in the synthesis of compounds and animal treatment, and to provide more detailed experimental steps and conditions (e.g., temperature, time, concentration, etc.).

The current experimental design mainly focuses on the measurement of biochemical and metabolic indexes, but lacks the analysis of tissue sections (e.g., pathological sections of liver, muscle, adipose tissue) and the detection of gene expression levels.

After supplementing the indicators of tissue sections, proteins and gene expression, a multilevel analysis is carried out in the discussion analysis, which can truly involve the biological effects of the complexes and their mechanisms of action.

The similarities and differences in mechanism and effect between this complex and other similar compounds should be discussed in detail to highlight the unique contribution of this study.

Add to the discussion an assessment of the potential clinical applications of the complex and suggestions for future research directions. For example, to explore the safety and efficacy of the complex in prolonged use and whether it can be used in combination with existing antidiabetic drugs to enhance efficacy.

It is recommended that the novelty of the study and the potential for clinical application be clearly identified in the conclusion.

The manuscript contains some grammatical errors and unclear sentence structure, some of which are listed in “* Comments on the Quality of English Language”. The authors are requested to check the manuscript again thoroughly.

Comments on the Quality of English Language

"Vanadium coordination complexes are more efficacious compared to salts, and one compound, bisethylmaltolatoxovanadium(IV) (BEOV) was in Phase 1 and Phase 2 clinical trials, at which point renal toxicity caused the compound to be abandoned."

"The effects of [VIVO(octd)] were evaluated in an animal model (female Swiss mice), whose insulin-resistant diabetes was triggered (promoted) by treatment with dexamethasone for 7 days, as well as being compared with groups treated with metformin."

"Glucocorticoids (GC) are steroid hormones synthesized and secreted by the adrenal cortex under the control of the hypothalamic-pituitary-adrenal axis, which connects to the glucocorticoid receptor (GR) regulating several physiological functions."

"Dexamethasone treatment provoked a reduction in body weight gain and food intake. The reduction in energy intake seems not to be the main cause for decreased body weight gain."

Author Response

Journal Pharmaceuticals (ISSN 1424-8247)

Manuscript ID pharmaceuticals-3027971

Title: Dexamethasone-induced insulin resistance attenuation by oral sulfur-oxidovanadium(IV) complex treatment in mice

Special Issue: Recent Advances and Future Approaches in Preventive and Therapeutic Effects of Vanadium Complexes

RESPONSES TO REVIEWERS

We want to thank the reviewers for the opportunity to improve our manuscript.  We have done our best to respond to the reviewers and, in the following, we describe the step-by-step changes we have made to our manuscript.  The changes made in the manuscript are described below for the convenience of the reviewers and the editor.

Reviewer comments:

REVIEWER 3 - Steps are needed to detail the reasons for the selection of the sulfur-oxovanadium(IV) complex and its advantages over existing treatments.

  1. Over-representation of references over 10 years.

Response: We have added some points to address the major critique about the interpretation of the results, especially about the novelty and its advantages over existing treatments. We, therefore, carefully address all suggestions made by the reviewers in the manuscript. We have included important key papers in the field that have been cited many times and are good reference material, with significant contributions to the vanadium area. Indeed, a lot of relevant work was done more than 10 years ago and should be used. For example, references [55-61] highlight the original method used in this article, and therefore the works responsible for the initial development of these methods were cited.

  1. In the “Materials and Methods” section, it is suggested to increase the detailed description of each step, especially in the synthesis of compounds and animal treatment, and to provide more detailed experimental steps and conditions (e.g., temperature, time, concentration, etc.).

Response: The section 4.2 was rewritten to address those suggestions.

  1. The current experimental design mainly focuses on the measurement of biochemical and metabolic indexes but lacks the analysis of tissue sections (e.g., pathological sections of liver, muscle, adipose tissue) and the detection of gene expression levels.

Response: Unfortunately, although your suggestion is valuable for future papers, we did not measure gene expression levels because the mechanistic study was not a focus of this manuscript. Our goal is to demonstrate in vivo antihyperglycemic effects of a known oxidovanadium compound in dexa-induced IR in mice. We feel that is a very important discovery as a pioneering study. The next step is to describe the possible cellular mechanisms involved, such as gene and protein expression in tissue metabolically essential such as liver, white adipose, and skeletal muscle. In the future, this idea will be considered.

  1. After supplementing the indicators of tissue sections, proteins and gene expression, a multilevel analysis is carried out in the discussion analysis, which can truly involve the biological effects of the complexes and their mechanisms of action.

Response: We plan to study the cell signaling pathways responsible for this compound's antihyperglycemic effect in the future.  

  1. The similarities and differences in mechanism and effect between this complex and other similar compounds should be discussed in detail to highlight the unique contribution of this study.

Response: There is difficulty in comparing the mechanisms and effects of the complex tested in this study with other complexes chemically like the S2O2 coordination mode since the signaling pathways are different. In addition, the articles indicated were tested on different animals and experimental models.

[1] H. Sakurai, H. Sano, T. Takino, H. Yasui, An orally active antidiabetic vanadyl complex, bis(1-oxy-2-pyridinethiolato)oxovanadium(IV), with VO(S2O2) coordination mode; in vitro and in vivo evaluations in rats, J. Inorg. Biochem. 80 (2000) 99–105, https://doi.org/10.1016/S0162-0134(00)00045-3.

[2] H. Sakurai, H. Sano, T. Takino, H. Yasui, A new type of orally active insulin mimetic vanadyl complex: bis(1-oxy-2-pyridinethiolato)oxovanadium(IV) with VO(S2O2) coordination mode, Chem. Lett. 28 (1999) 913–914, https://doi.org/10.1246/cl.1999.913.

[3] A. Katoh, M. Yamaguchi, R. Saito, Y. Adachi, H. Sakurai, Insulinomimetic vanadylhydroxythiazolethione complexes with VO(S2O2) coordination mode: the correlation between the activity and Hammett’s substituent constant, Chem. Lett. 33 (2004) 1274–1275, https://doi.org/10.1246/cl.2004.1.

[4] I. Osi´nska-Kr´olicka, H. Podsiadły, K. Bukiety´nska, M. Zemanek-Zboch, D. Nowak, K. Suchoszek-Łukaniuk, M. Malicka-Błaszkiewicz, Vanadium(III) complexes with L-cysteine - stability, speciation and the effect on actin in hepatoma Morris 5123 cells, J. Inorg. Biochem. 98 (2004) 2087–2098, https://doi.org/10.1016/j.jinorgbio.2004.09.013.

  1. Add to the discussion an assessment of the potential clinical applications of the complex and suggestions for future research directions. For example, to explore the safety and efficacy of the complex in prolonged use and whether it can be used in combination with existing antidiabetic drugs to enhance efficacy.

Response: We have included this suggestion in the manuscript (page 14, lines 564-567). In future work, we will evaluate the molecular level of proteins linked to the insulin pathway, such as IR, IRS1, AKT, and glucocorticoid receptor (GR) to propose which metabolic pathways are activated by this vanadium complex.

  1. It is recommended that the novelty of the study and the potential for clinical application be clearly identified in the conclusion.

Response: We have revised the current version. The results suggest that oxovanadium complex has emerged as an efficient therapeutic proposal capable of controlling metabolic side effects associated with the therapeutic use of synthetic glucocorticoids. This becomes particularly important for patients who need to use dexamethasone, such as transplant patients, and more recently, those with severe acute respiratory syndrome (SARS) in coronavirus-infected patients. We have included it in the conclusion section (page 15, lines 556-564).

The manuscript contains some grammatical errors and unclear sentence structure, some of which are listed in “* Comments on the Quality of English Language”. The authors are requested to check the manuscript again thoroughly. Comments on the Quality of English Language

"Vanadium coordination complexes are more efficacious compared to salts, and one compound, bisethylmaltolatoxovanadium(IV) (BEOV) was in Phase 1 and Phase 2 clinical trials, at which point renal toxicity caused the compound to be abandoned."

"The effects of [VIVO(octd)] were evaluated in an animal model (female Swiss mice), whose insulin-resistant diabetes was triggered (promoted) by treatment with dexamethasone for 7 days, as well as being compared with groups treated with metformin."

"Glucocorticoids (GC) are steroid hormones synthesized and secreted by the adrenal cortex under the control of the hypothalamic-pituitary-adrenal axis, which connects to the glucocorticoid receptor (GR) regulating several physiological functions."

"Dexamethasone treatment provoked a reduction in body weight gain and food intake. The reduction in energy intake seems not to be the main cause for decreased body weight gain."

Response: We have carefully revised the English Language throughout the manuscript. The grammatical errors and sentence structures have been carefully checked by the authors.

Round 2

Reviewer 2 Report

Comments and Suggestions for Authors

The author has addressed all the raised issues, and I am satisfied with his resposes. 

Reviewer 3 Report

Comments and Suggestions for Authors

1. Aiming to determine the role of oxovanadium compounds in dexamethasone-induced insulin-resistant mice at the histological level, increased analysis of pathological sections of liver, muscle and adipose tissue is necessary. By methods such as H&E staining and immunohistochemistry, tissue structural changes and cellular lesions can be visualized.

2. Although the focus of the current study is on anti-hyperglycemic effects, the mechanism of action of oxovanadium compounds must be initially explored, and the addition of experiments such as RT-qPCR and Western blot to detect changes in the expression of key genes and proteins is necessary. At least indirect evidence obtained from RT-qPCR experimental data is needed.